# GUESS & SKETCH: LANGUAGE MODEL GUIDED TRANSPILATION

**Celine Lee**[♠]   **Abdulrahman Mahmoud**[†]   **Michal Kurek**[†]   **Simone Campanoni**[♡]
**David Brooks**[†]   **Stephen Chong**[†]   **Gu-Yeon Wei**[†]   **Alexander M. Rush**[♠]
[♠] Cornell University,  [†] Harvard University  [♡] Northwestern University
cl923@cornell.edu

## ABSTRACT

Maintaining legacy software requires many software and systems engineering hours. Assembly code programs, which demand low-level control over the computer machine state and have no variable names, are particularly difficult for humans to analyze. Existing conventional program translators guarantee correctness, but are hand-engineered for the source and target programming languages in question. Learned transpilation, i.e. automatic translation of code, offers an alternative to manual re-writing and engineering efforts. Automated symbolic program translation approaches guarantee correctness but struggle to scale to longer programs due to the exponentially large search space. Their rigid rule-based systems also limit their expressivity, so they can only reason about a reduced space of programs. Probabilistic neural language models (LMs) produce plausible outputs for every input, but do so at the cost of guaranteed correctness. In this work, we leverage the strengths of LMs and symbolic solvers in a neurosymbolic approach to learned transpilation for assembly code. Assembly code is an appropriate setting for a neurosymbolic approach, since assembly code can be divided into shorter non-branching basic blocks amenable to the use of symbolic methods. GUESS & SKETCH extracts alignment and confidence information from features of the LM then passes it to a symbolic solver to resolve semantic equivalence of the transpilation input and output. We test GUESS & SKETCH on three different test sets of assembly transpilation tasks, varying in difficulty, and show that it successfully transpiles 57.6% more examples than GPT-4 and 39.6% more examples than an engineered transpiler. We also share a training and evaluation dataset for this task.

## 1   INTRODUCTION

The increasingly heterogeneous landscape of hardware architectures and their instruction set architectures (ISAs) marks a large and growing need to develop support for cross-ISA software management. This challenge is especially relevant for hardware-specific legacy software which must be re-written to run on any other hardware. Many high-usage source code files also contain in-lined assembly code, which requires porting to alternate hardware architectures. Automated cross-ISA software support has been of interest in the computer architecture community for decades (Armengol-Estapé et al., 2023; Wang et al., 2018; Bellard, 2005; Ardestani & Renau, 2013; Sanchez & Kozyrakis, 2013). Emulators, virtual machines, and containerized applications allow users to run software on different host hardware by simulating the architecture of the hardware platform that the software is compiled for. However, this option can be unwieldy and compute-inefficient. Assembly-to-assembly *transpilation* [1] (Ami; occ, 1989), the process of automatically porting software from one ISA to another, offers a way to generate software that can be natively executed on the new hardware. However, current transpilation tools are engineered for the specific source and target hardware architecture, so they scale poorly as new ISAs are introduced.

Neural machine learning techniques are a natural fit for transpilation. Assembly program translation pairs can be generated by cross-compiling C or C++ programs using different existing compilers and

---

[1] *"Transpiler"* describes the general code translation task that our method targets, but we note that the focus of this paper is assembly-to-assembly transpilation.

compiler flags, providing vast amounts of training data. Pairs have the same semantics since they originate from the same high-level program. Assembly code syntax is rigid but simple compared to natural language and most high-level programming languages, settings that existing language models have been shown to perform well in (Devlin et al., 2019; Feng et al., 2020; Radford & Sutskever, 2018; Lewis et al., 2019; Chen et al., 2021). Evaluation in this setting can also be done automatically by comparing execution of the input code and the resulting code.

However, a key weakness of language models in this setting is their inability to perform long-tail logical reasoning (Kandpal et al., 2022; Miceli-Barone et al., 2023). Assembly code transpilation requires reasoning about the complex semantics of programs. Additionally, specific challenging phenomena, such as differing implementations of mathematical operations on different ISAs, are critical and arise frequently in assembly code.

Motivated by the symbolic properties of logical reasoning in the problem of transpilation, we propose a neurosymbolic method to transpilation. Purely symbolic methods are built on correctness guarantees, but generally can only handle short programs before encountering computational intractability. Classical synthesis techniques struggle to scale past $\sim 6$ lines of assembly code (Hu et al., 2023). Purely neural language modeling approaches are powerful general translators but have critical failure points that cause program breakdown. We argue for the value of a mixed-method, i.e. neurosymbolic, approach that uses probabilistic language models to obtain helpful information for transpilation, then passes such information to an ISA semantics-aware solver to complete the transpilation process.

Our method, GUESS & SKETCH, uses core properties from the language model to extract symbolic methods for transpilation. During the neural GUESS phase, a trained language model produces candidate translations for a given input, identifies potential errors in the output, and extracts semantically-aligned subsequences from the input and output sequences. Potentially erroneous aligned subsequences are passed to the symbolic SKETCH phase, where the input subsequence is used as a specification to correct the output subsequence.

We demonstrate the feasibility of our method by porting assembly programs from ARMv8 to RISC-V and vice-versa, but note that our method can generalize to various source and target languages. In order to test our method, we introduce a new benchmark consisting of 3 transpilation problems varying in difficulty and domain. We identify weaknesses in engineered symbolic approaches to the task. We also find that existing neural network approaches, using both fine-tuned and pre-trained off-the-shelf large language models, struggle with transpilation. In contrast, our method combines the strengths of both neural and symbolic approaches and successfully transpiles 57.6% more examples than GPT-4, 39.6% more examples than an engineered transpiler, and 13.2% more examples than the most competitive baseline.

## 2   RELATED WORK

**Learned code translation.**   Code transpilers (or transpilers) translate from one programming language to another. The core challenge in this space is preserving operational semantics across the source and target language, while operating within the strict syntax and vocabulary of both. One approach to this task is to train neural machine translation systems with paired code sequences for the task, such as language model (Lewis et al., 2019) or tree-to-tree neural networks (Chen et al., 2018). Approaches such as Transcoder (Roziere et al., 2020) have also presented an unsupervised approach to neural source code-to-source code translation, in which they only require monolingual training data and take advantage of three training objectives: cross-lingual masked language modeling, denoising auto-encoding, and back-translation. Follow-up works use the LLVM intermediate representation (Roziere et al., 2022) and automatically-generated unit tests (Szafraniec et al., 2023) to further improve this approach. Older statistical approaches have mined parallel code from repositories and generated grammar-based statistical machine translation models (Nguyen et al., 2013; Karaivanov et al., 2014; Koehn et al., 2007). These outputs of these prior learned approaches are the generation directly extracted from the model. GUESS & SKETCH instead incorporates knowledge of the semantics of the source and target languages in a symbolic solver that improves semantic correctness the produced output. Additionally, as far as we are aware, we are the first to present a learned approach for learning assembly translation, a lower-level programming language than higher-level programming languages such as Python, Java, and even C.

**Emulators and engineered transpilers.** Executing code on a platform different than the one for which it was created is a long-desired task. Apple's Rosetta (app) software was designed to ease the transition of applications between hardwares by automatically translating binary executables from the previously supported to the new ISA. Specifically, Rosetta in 2006 supported the transition from PowerPC to Intel x86 processors. Rosetta 2 released in 2020 enabled translation from x86-64 based processors to support by ARM64-based Apple silicon. Emulators and virtualizers allow users to execute code designed for another target hardware by simulating the target hardware ISA atop the host hardware. QEMU (Bellard, 2005) is one popular emulator and virtualizer that can emulate various architectures on certain host architectures. Other assembly transpilers have been written to translate assembly from one language to another, such as from ARM to RISC-V (Schorr et al., 2020). However, these emulators and transpilers take years to develop. GUESS & SKETCH, on the other hand, leverages the translation abilities of a learned model to perform a bulk of the transpilation.

**Neurosymbolic program synthesis.** Program synthesis is the task of generating computer programs according to some correctness specification (Lee et al., 2021). In the context of program translation, the correctness specification is the semantics of the input program itself. We discuss here some works that take a combined neural and symbolic approach to the program synthesis task, similar to our own approach. Nye et al. (2019) train an LSTM-based model to generate program sketches from some input specification, then use the generated sketch and specification to search for a satisfying program. Guo et al. (2022) devise a top-down grammar-based method to selectively expand nonterminals in a program syntax tree. The incomplete program tree is converted to a sketch that is passed to the symbolic sketch solver to generate a full program. Unlike these previous works, our method infers the sketch using attributes of a single autoregressive language model. The benefit of our approach is over directly producing the sketch or generating based on a grammar is that we avoid encoding specific sketch and language technicalities into the training process.

## 3 BACKGROUND

### 3.1 TRANSPILATION

The task of transpilation is to take an input program $P_x$, represented as sequence of tokens $x$, and produce the semantically-equivalent program $P_y$ represented as sequence of tokens $y$. Let $\mathcal{D}$ be the domain of all program inputs. For simplicity we represent programs as functions that map inputs to a deterministic measurable output, either an integer or program failure: $P_* : \mathcal{D} \to (\mathbb{Z} \cup \bot)$. Semantic equivalence can be measured by checking that for all inputs in $\mathcal{D}$, both programs produce the same execution outputs: $x \equiv y : \forall d \in \mathcal{D} : P_x(d) = P_y(d)$. In practice, we test the full programs on a feasible subset of $\mathcal{D}$ determined by the objective of the source program.

When working with programs, we will also assume we can partition the tokens into $\mathcal{B}_x$ non-overlapping subsequences $x = x_{b_1}, \ldots, x_{b_{|\mathcal{B}_x|}}$ where each $b \in \mathcal{B}_x$ defines a span over $x$. Sub-sequences are defined so that they can individually be converted to programs $P_{x_b}$. Details for identifying such subsequences for assembly and translating them into a program representation conducive for symbolic reasoning in a sketch solver are shared in Appendix A.1.

### 3.2 GENERATIVE LANGUAGE MODELS

Let $(x, y) \in (\mathcal{V}^L, \mathcal{V}^L)$ denote an input and output sequence pair where $\mathcal{V}$ is the shared vocabulary of tokens and $L$ is the maximum length. The objective of a (conditional) generative language model is to autoregressively produce the correct output $y$ from input $x$:

$$\underset{y \in \mathcal{V}^L}{\arg\max} \prod_t p(y_t | y_{<t}, x)$$

Modern language models are based on the Transformer architecture (Vaswani et al., 2017). Transformers use attention (Parikh et al., 2016), a routing mechanism that provides a distribution over the input tokens used for predicting the next word. Intuitively, attention learns to indicate which part of the input to weigh more for each output. We can extract the model's attention between the input

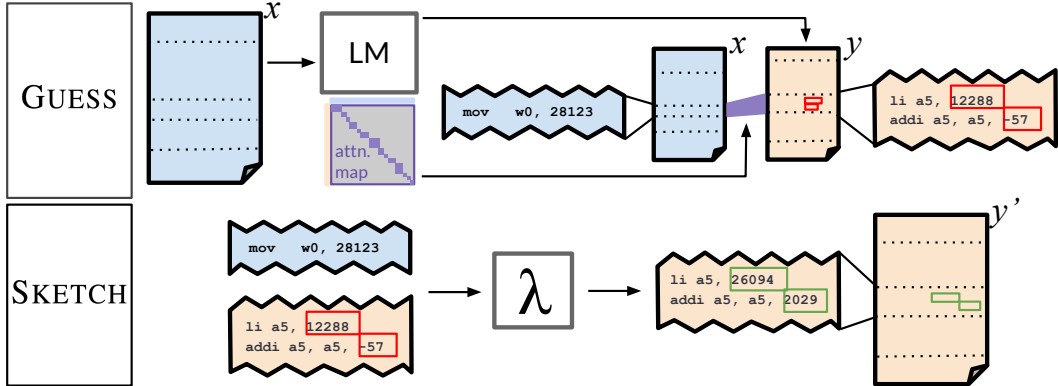

Figure 1: In the GUESS (top) phase, the full input sequence $x$ (blue) is passed to a trained language model (LM), which produces a candidate translation $y$ (orange), identifies potential mistakes (red), and extracts subsequence alignment (purple) from attention between the input and output (attn. map). In the SKETCH (bottom) phase, aligned input and output subsequences are passed to a symbolic solver $\lambda$ to correct errors identified in the GUESS phase. The final output $y'$ is constructed by recombining corrected subsequences.

sequence $x$ and output sequence $y$ as a series of stochastic matrices at each layer mapping every output index to a probability distribution over input indices[2]: $M \in \Delta^{|y| \times |x|}$.

### 3.3 SKETCHING

Sketching (Solar-Lezama, 2009; Solar-Lezama et al., 2006a) is an approach to program synthesis in which a *partial program* outlines the high-level implementation, then a synthesizer populates the omitted low-level details by ensuring that the resulting code passes some given correctness specification. Partial programs are expressed in a procedural programming language augmented with a single added construct: a symbolic constant expressed as a hole, denoted •. Programs expressed in this form, with holes as placeholders for concrete values, are *sketches*. In our notation, the partial program sequence is composed of tokens from the vocabulary and an added hole token: $\mathcal{S} = (\mathcal{V} \cup \{\bullet\})^*$. Program sequences $x$ are compiled by a semantics-aware translator into representations $P_x$ in the procedural programming language understandable by the solver.

The correctness specification is set by source program $P_x$. The goal of the synthesizer is to identify the mapping $\phi : \mathcal{S} \to \mathcal{V}^*$ that populates the holes of the partial program sequence $s$ to produce the full program sequence $\phi(s)$ whose corresponding program is semantically equivalent to the source program: $\forall d \in \mathcal{D} : P_{\phi(s)}(d) = P_x(d)$.

The synthesis engine reduces the resulting programmatic sketch representation to a constraint satisfaction problem solved using counterexample guided inductive synthesis (Solar-Lezama et al., 2006b) to find values for the holes.

## 4 NEUROSYMBOLIC TRANSPILATION: GUESS & SKETCH

Given an input program $P_x$ represented as sequence $x \in \mathcal{V}^L$, our goal is to learn to generate a semantically-equivalent output sequence $y \in \mathcal{V}^L$ which represents program $P_y : P_x \equiv P_y$. Programs are comprised of function definitions that are generally independent from one another, so functions are individually translated then stitched back together. See details in Appendix A.

The challenge of our neurosymbolic approach is that language models operate on prefixes, performing inference by producing one token at a time, while sketch-based methods reason with partially complete sequences. **To meaningfully pass information between the language model and the symbolic solver, we must extract relevant sequence-level information from the language model**

---

[2]In encoder-decoder models this comes from cross-attention, for decoder-only models by renormalizing self-attention.

**for the solver to reason over with.** Specifically, the solver needs candidate output translations and their semantic alignment in the input.

Our method breaks the problem into stages that can be better solved by the complementary strengths of neural and symbolic methods: a probabilistic machine learning language model produces candidate translations, then alignment and confidence information is extracted and passed to a semantics-aware solver to filter the search spaces for a correct solution. The pipeline for the GUESS & SKETCH approach is illustrated in Figure 1.

## 4.1 GUESS: STRUCTURED CANDIDATES FROM A GENERATIVE MODEL

The GUESS phase produces guesses as tuples. For an input sequence $x$, GUESS produces tuples composed of: a candidate transpilation $y$, alignments between subsequences: $A \in \mathcal{B}_x^{|\mathcal{B}_y|}$, and potential token-level errors in the prediction: $E \in \{0, 1\}^{|y|}$.

**Candidates.** To produce candidate sequences we follow a standard generative approach. We first train a generative language model on paired source language and target language program sequences. Once trained, candidate transpilations are produced by querying the model:

$$y \in \mathop{\text{top k}}_{y \in \mathcal{V}^L} p(y|x) \tag{1}$$

**Alignment.** Since the input and target output sequences are intended to be globally semantically equivalent, we assume output sequences locally align to input sequences. While there is not a one-to-one equivalence between tokens, subsequences of the two programs can be matched. We use this subsequence matching and the transformer attention to determine the alignment used by the sketch system. A sample extracted alignment matrix, along with the truth alignment matrix, is shown in Figure 2.

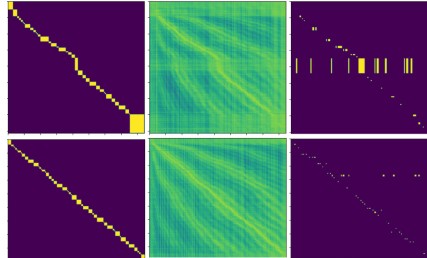

Figure 2: True subsequence alignment (l), attention (r), and projected subsequence alignment (r) from the GUESS model.

Alignment is represented as a vector between subsequences: $A$. To extract the alignment from the language model, we average the transformer attention matrices connecting $x$ and $y$ at single layer to form a stochastic matrix $M \in \Delta^{|y| \times |x|}$. We then set the alignment $A_{b_j} = b_i$ for the input subsequence with the highest aggregate attention score. Aggregate attention score is given by norm of the submatrices i.e. $\forall b_{j'} \in \mathcal{B}_x : \|M_{b_j, b_i}\| \geq \|M_{b_{j'}, b_i}\|$.

**Guesses and Errors.** The generative model is also used to identify tokens where it is most likely guessing. First we check if the output token $j$ is predicted with probability less than some value $\gamma$:

$$p(y_j|y_{<j}, x) < \gamma \tag{2}$$

These low-confidence prediction points correlate to long-tail code phenomena, i.e. instances that arise rarely in the data distribution, and are where the model may have made a translation mistake. The second case is if the general model is confident, but the program violates a domain specific heuristic, specifically if the token or its aligned input subsequence reference some entity not described in scope. If either of these conditions are satisfied, the tokens in question are marked as potentially erroneous: $E \in \{0, 1\}^{|y|}$.

## 4.2 SKETCH: REASON OVER ALIGNED CANDIDATES

The SKETCH phase produces a full synthesized transpilation using symbolic program solver methods and information from the GUESS phase. Note that determining full program equivalence is an undecidable problem, so we focus on solving for errors in individual subsequences $\mathcal{B}_y$.

**Create the sketch.** We create a sketch $s$ for each subsequence $b \in \mathcal{B}_y$ that has an possible error from the first stage. The sketch is created from $y_b$ by replacing each position in $j \in b$ that also

**Algorithm 1** GUESS & SKETCH Pseudocode

---

**procedure** GUESS & SKETCH(x)
    **for** $y, A, E \in \text{GUESS}(x)$ **do**             ▷ produce candidates, alignments, potential errors
        **for** $b$ in $\mathcal{B}_y$ **do**
            **if** $P_y \equiv P_x$ **then return** $y$
            **if** $E_j$ for any $j \in b$ **then**             ▷ identify potential error
                $b_x \leftarrow A_b$             ▷ get aligned input index
                $s \leftarrow \text{PLACE\_HOLES}(y_b, E)$             ▷ produce sketch sequence
                $\phi \leftarrow \arg\max_\phi \mathbb{1}(P_{x_{b_x}} \equiv P_{\phi(s)})$           ▷ solve for solution (synthesizer)
                **if** $\phi$ success **then**
                    $y \leftarrow \text{UPDATE}(b, \phi(s))$             ▷ update subseq.

---

satisfies $E_j \neq 0$ with a hole •, i.e. potentially erroneous tokensn . The correctness specification is set by the program represented by the aligned input subsequence $x_{b_x}$ where $A_b = b_x$. Correctness specifications must be based on complete semantics, so for input subsequences with out-of-scope references, we extract the definition of the referenced entity from the full program. The retrieved entity definition is used to complete the semantics of the correctness specification.

A semantics-aware translator lifts the sketch and correctness specifications into their sketch solver programmatic representations $P_s$ and $P_{x_{b_x}}$, respectively. Details about this translation process for our assembly language experiments are shared in Appendix A.1.

**Solve the sketch.** To solve the sketch is to find a mapping $\phi$ that correctly populates all holes of the partial program sequence $s$ to satisfy the correctness specification: $\forall d \in \mathcal{D} : P_{x_{b_x}}(d) = P_{\phi(s)}(d)$.

If a solution populating all holes of the partial program sequences is found by the sketch solver, it is applied to $s$ and the updated subsequence $\phi(s)$ replaces the subsequence in the full program sequence. If the subsequence had an out-of-scope reference, the solver would have also resolved a definition of the referenced entity. The resolved referenced entity definition is also updated in the full program. In cases where a sketching solution cannot be found, GUESS & SKETCH resorts to the original prediction. Since our method always at least defaults to the original generation, the correctness of GUESS & SKETCH is lower-bounded by the correctness of the initial guess. This full process is summarized in Algorithm 1.

## 5 EXPERIMENTAL SETUP

**Dataset** Our experiments focus on transpilation between real programs compiled to different ISAs, specifically the ARMv8 and RISC-V assembly languages. ARMv8 and RISC-V are both reduced instruction set architectures (ISAs), and have some similarities in instructions (Hennessy & Patterson, 2011). We construct training and evaluation datasets for this task.

Training data is composed of 307,916 ARMv8 and RISC-V assembly file pairs compiled from C code files from The Stack (Kocetkov et al., 2022). All selected source C files can be independently compiled to assembly using the standard C libraries (e.g. stdlib, stdio). The C files are compiled to both ARMv8 and RISC-V target architecture assembly files under the `-O0`, `-O1`, `-O2`, and `-O3` optimization flags using cross-compilers `aarch64-linux-gnu-gcc` and

| Test Dataset | # | Avg len | In | Out |
|---|---|---|---|---|
| Unix Commands | 11 | 96 | ✓ | ✓ |
| Project Euler | 45 | 159 | | ✓ |
| Benchmarks | 16 | 484 | ✓ | ✓ |

Figure 3: Test sets for transpilation. Length is measured as number of lines in the assembly file, and is averaged across both ARMv8 and RISC-V architectures under the `-O0` optimization flag.

`riscv64-linux-gu-gcc`. The resulting dataset is shared on HuggingFace[3].

Inference of the system is evaluated on 3 different test sets, summarized in Table 3. Code is emulated in Docker images with QEMU Bellard (2005). *Project Euler* is constructed from 45 C implementa-

---

[3]https://huggingface.co/datasets/celinelee/paired_arm_risc

| Method | RISC-V to ARMv8 | | | ARMv8 to RISC-V | | |
|---|---|---|---|---|---|---|
| | Proj. Euler | Benchmx | Unix Cmds | Proj. Euler | Benchmx | Unix Cmds |
| Few-shot (GPT4) | 11.1% | 0 | 18.2% | 4.44% | 0 | 27.3% |
| Transpiler | - | - | - | 24.4% | 12.5% | 54.5% |
| FT StarCoder | 8.9% | 0 | 36.4% | 8.9% | 0 | 36.4% |
| FT CodeLLaMA | 11.1% | 0 | 36.4% | 2.2% | 0 | 36.4% |
| Encoder-Decoder | 68.9% | 6.3% | 36.4% | 66.7% | 6.25% | **81.2%** |
| GUESS & SKETCH | **80%** | **18.8%** | **81.2%** | **75.6%** | **25.0%** | **81.2%** |

Table 1: Main Transpilation results on full program accuracy (Project Euler, Benchmarks, and Unix Commands test sets). Bold shows best results with $p < 0.01$ significance.

tions of Project Euler mathematical challenge problems[4]. *Benchmarks* is 16 C implementations of programs in The Computer Language 23.03 Benchmarks Game[5]. *Unix Commands* is 11 C implementations of Basic Unix commands[6].

For verification, all test sets are cross-compiled to the ARMv8 and RISC-V architectures under the `-O0` flag. System performance is measured by execution output match. We sample the top 100 candidate guesses for a given full assembly file.

**System** We experiment with two different types of generative language models: a smaller transformer encoder-decoder model with a bidirectional encoder and autoregressive decoder based on the BART architecture (Lewis et al., 2019), and a larger transformer decoder-only models pre-trained on code (Li et al., 2023; Rozière et al., 2023). The first model class is trained from scratch where the second is pretrained. All language models are trained on one NVIDIA RTX A6000 GPU. The encoder-decoder models are trained for 156 hours total and the pre-trained decoder-only models are fine-tuned for 240 hours total. Pre-trained models are fine-tuned with LoRA (Hu et al., 2022). Details of training are shown in Table 4. All resulting models are shared on Huggingface [7]. We use confidence threshold $\gamma = 0.9$, although we found that it was not critical for accuracy. Additional $\gamma$ experiments are in Appendix A.

The symbolic solver is built with Rosette (Torlak & Bodik, 2013), a programming language for synthesis and verification built on top of the Z3 (de Moura & Bjørner, 2008) SMT solver. The input space is restricted to 16-bit bitvectors, consistent with the register sizes of the ARMv8 and RISC-V architectures used.

**Baselines** We consider several alternate approaches to assembly transpilation. With *Few-shot learning* (Brown et al., 2020), we prompt GPT-4 (OpenAI, 2023) with instructions and a couple of examplar input-output assembly pairs to obtain a transpilation for a given input assembly file. See details of the specific prompt in Appendix D.1. *Transpiler*s are manually-engineered transpilers that convert the given source assembly to the given target assembly. These are programmatically written for the specified source-to-target-hardware, so for source-target hardware pairs for which we cannot find a transpiler, we cannot obtain numbers for this baseline. We use the engineered ArmV8-to-RISCV64 transpiler written by members of the IBM Research Haifa team [8]. We did not find a transpiler from RISC-V to ARMv8. LM only methods, *FT StarCoder* (Li et al., 2023), *FT CodeLLaMA* (Rozière et al., 2023), *Encoder-Decoder* (Lewis et al., 2019), are the purely neural approaches to machine translation, in which we train or fine-tune a language model with the paired assembly data. The *Encoder-Decoder* method is equivalent to just the GUESS method of our approach.

# 6 RESULTS AND ANALYSIS

Performance of our methods on the test sets are shown in Table 1. GUESS & SKETCH outperforms all alternative approaches with $0.01$ significance level [9]. The Few-shot approach, even with

---

[4]https://github.com/eagletmt/project-euler-c

[5]https://benchmarksgame-team.pages.debian.net/benchmarksgame/index.html

[6]https://github.com/yadu007/Basic-Unix-Commands-Implementation

[7]https://huggingface.co/celinelee/bartlarge_{armtorisc/risctoarm}_cloze2048

[8]https://github.com/schorrm/arm2riscv

[9]According to a two sample z-test.

| | | Few-shot | Starcoder | CodeLlama | Transpiler | Enc-Dec | GUESS & SKETCH |
|---|---|---|---|---|---|---|---|
| Process | Length | 2 | 7 | 7 | 0 | 6 | 6 |
| | Failure | 0 | 0 | 0 | 34 | 0 | 0 |
| Compile | ISA | 62 | 50 | 57 | 0 | 2 | 2 |
| | References | 3 | 5 | 5 | 0 | 11 | 1 |
| Semantics | Copying | 0 | 0 | 0 | 0 | 1 | 1 |
| | Logic | 1 | 5 | 3 | 0 | 3 | 3 |
| | Memory | 10 | 10 | 9 | 0 | 2 | 2 |
| | Math | 7 | 3 | 0 | 0 | 2 | 3 |
| Correct | | 5 | 10 | 6 | 11 | 61 | 70 |

Table 2: Analysis of failures by different transpilation methods. Collected on the Project Euler test set. Categories are listed in order of bottleneck precedence.

the largest existing language model today, GPT-4, cannot successfully perform most transpilations. GUESS & SKETCH even outperforms the engineered Transpiler, which fails to translate programs for which it cannot recognize even one instruction. We run several GUESS-only models, comparing from-scratch training to pre-trained models. Interestingly, the fine-tuned pre-trained large language models perform much worse than even just the trained smaller encoder-decoder model. The best-performing baselines is the Encoder-Decoder approach, which we use for the full GUESS & SKETCH. Further experiments testing the performance gain of GUESS & SKETCH over the Encoder-Decoder approach on more test programs are shared in Appendix B, and support the same 10% increase in correct transpilations.

**Error Analysis** Table 2 classifies assembly transpilation errors under one of several categories, determined by bottleneck failure reason: mathematic, copying, ISA, references, logic, memory, and length. See descriptions of each in Appendix C and examples in Appendix C.1.

The encoder-decoder model (GUESS) makes few ISA mistakes, but runs into a number of errors in semantics and out-of-scope references, some of which are resolved by the solver in GUESS & SKETCH. However, unless the semantics of all of its erroneous subsequences are resolved, an incorrect transpilation is not corrected. That is, even though mathematically erroneous subsequences are being resolved across the examples in the test sets, if the bottleneck problem is not resolved or not all errors are properly aligned and solved, the transpilation still fails.

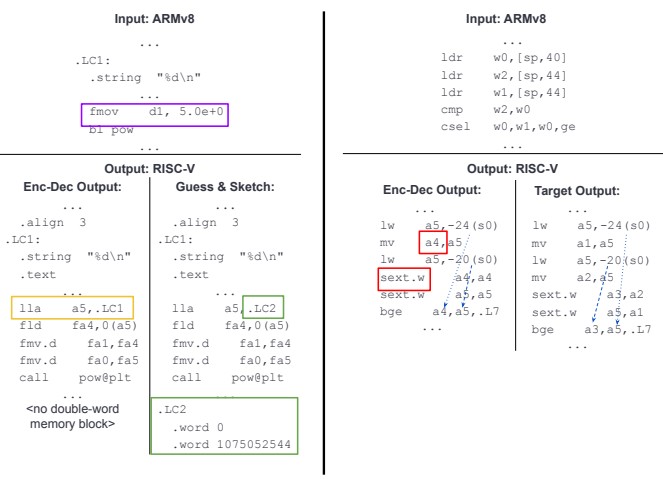

Figure 4: Example outputs.

Interestingly the other approaches fail to transpile or compile before even reaching semantics. For few-shot, the model generates invalid instructions, despite the prompt including a translation instructions as well as multiple exemplar transpilations. Fine-tuning models generate invalid assembly from pretraining despite the fine-tuning phase. On the other hand, the manually engineered transpiler is unable to process many examples at all.

Figure 4 shows two example outputs. The left shows a guess that is resolved. The language model output (bottom, left) predicts tokens for the incorrect global memory reference, highlighted in yel-

|  | Project Euler | |
|  | RISC-V to ARMv8 | ARMv8 to RISC-V |
| --- | --- | --- |
| Encoder-Decoder | 30.1 | 34.3 |
| GUESS & SKETCH | **21.3** | **25.3** |

Table 3: Average number of samples used by the encoder-decoder and GUESS & SKETCH approaches for the Project Euler test set. The range is $[1, 100]$. (Lower is better.)

low. According to the model cross-attention, these tokens most align to those of the corresponding `fmov` instruction in the input assembly (top), highlighted in purple. However, in the predicted full assembly program, no memory location is produced with the double-word IEEE representation for the desired float `5.0e+0`. After resolution with GUESS & SKETCH, a correct memory location is generated and the memory reference is updated (bottom, right), highlighted in green. The example on the right shows a problem that GUESS & SKETCH does not resolve. The LM output (bottom, left) predicts tokens for the register values with low confidence, highlighted in red. A correct solution is shown (bottom, right). The register use and logic flow is inconsistent.

**Sampling**    Aside from solving more examples in the test dataset, GUESS & SKETCH also reduces the number of samples needed from the underlying LM. For a set of test examples, they are correctly transpiled using the encoder-decoder approach only after sufficiently many samples. Using GUESS & SKETCH, a handful of these are successfully transpiled with fewer samples. Table 3 shows the average number of samples from the LM used by the encoder-decoder approach and the GUESS & SKETCH approach during evaluation of the Project Euler test set. Examples that achieve a correct transpilation after the $k^{th}$ sample are logged to use $k$ samples, and examples that do not achieve a correct transpilation within 100 samples use 100 samples.

## 7    LIMITATIONS

While GUESS & SKETCH is significantly more effective than the baseline approaches, there are still several remaining open challenges.

- The SKETCH method is dependent on alignment with the source sequence. If GUESS fails to provide an accurate alignment than the sketch may be unable to correct the output issue.

- Memory management issues are hard for the sketch solver. These include reasoning about values on the stack at any given point in the program, register choice decisions that are incorrectly propagated during autoregressive generation, and loading memory addresses into the register.

- The best performing model is a mid-size encoder-decoder, which is strong at pattern matching, but likely cannot perform programmatic reasoning. Potentially larger code models could better solve some of the symbolic transpilation issues, if instruction hallucinations could be reduced.

- GUESS & SKETCH is limited in length by the context length of generative language models. Using convolutional methods such as SLeD (Ivgi et al., 2022) could resolve these mistakes in practice.

- We have no formal proof of equivalence, only checking on a small finite set of inputs.

## 8    CONCLUSION

In this work, we present GUESS & SKETCH, a neurosymbolic approach to assembly-to-assembly transpilation. GUESS & SKETCH extracts alignment and confidence information from a language model to guide a symbolic solver. We demonstrate the efficacy of this approach on three different test sets of assembly programs in the ARMv8 and RISC-V architectures. Future work to build on this approach is to identify and use patterns in the decoder attention of the language model that may be helpful for the solver, such as live variable analysis (Aho et al., 2006) patterns. Other future work may include transpiling to or from higher levels of code optimization and devising a mechanism to reason about more elements of the machine state, such as values on the stack.

## ACKNOWLEDGMENTS

We thank Justin Chiu, Amrit Baveja, Hao Tang, Yair Schiff, Omer Gul, Kevin Ellis, Ameesh Shah, Sahil Bhatia, and Adwait Godbole for helpful conversations and feedback throughout the project. This work is supported in part by the National Science Foundation (NSF) grant CCF-1704834. CL and AMR are sponsored by NSF Grant DRL-2229873 and 2037519.

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

## A  Implementation Details of Guess & Sketch for Assembly

**Function boundaries.** The length of assembly files often well exceeds the context window size of the language model. To handle this issue, we perform translation through the language model by separating functions from one another and translating them individually. This decision is grounded in the fact that for the ISAs tested, most information in the functions is independent of instructions in other functions. This is especially true with regard to the general structure of the computations rather than specific low-level details and values. The language models are trained on these separated assembly functions. In inference, the models are passed separated assembly functions, and the resulting function translations are concatenated back together to compose the full assembly program.

**Confidence threshold:** $\gamma$    The underlying language model can be very confident about incorrect predictions (Johnson et al., 2023; Vasconcelos et al., 2023). In the assembly translation setting, this often happens for example when referencing out-of-scope entities, as described at the end of Section 4.1. This is why domain specific heuristics can help the GUESS & SKETCH system identify which basic blocks to correct. To evaluate the effect of $\gamma$ on system performance, we sweep $\gamma = [0.8, 0.9, 0.95]$ with the Project Euler test set. A lower $\gamma$ would flag fewer potential errors for correction, which may reduce or maintain the number of instances of sketching. Fewer sketching instances may result in fewer corrections, but has the benefit of reduced computation time. We find that across these $\gamma$ values, the number of corrected programs is the same, but the inference runtime increases with $\gamma$. From $0.8$ to $0.95$, the inference time increases by 2.2x.

## A.1    ALIGNED SEQUENCES IN ASSEMBLY: PURE BASIC BLOCKS

Assembly basic blocks are sequences of code lines that have a single entry point and single exit point. That is, there are no branching operations within the code sequence (Patterson & Hennessy, 1990). We introduce *pure basic blocks*, a subset of basic blocks defined as sequences of assembly code lines that have a single entry point, a single exit point, and no memory or stack management within the code sequence. This constrains pure basic blocks to be code sequences in which all data is either passed in via values already loaded into registers, or constant values coded into the sequence. This decision to remove memory operations and other control flow instructions greatly simplifies the equivalence relation between source and target subsequences.

**Identifying out-of-scope references.**    In the context of assembly, out-of-scope references as potential mistakes are classified as any piece of code that use or reference global memory. Examples include the $lla$ instruction in the RISC-V architecture or custom string or function definitions.

**Extract pure basic blocks.**    From a given token in the sequence, we identify the surrounding pure basic block by inspecting the neighboring assembly lines. We greedily search lines upward and downward from the given token until one matches a section boundary definition, branching, memory management, or stack management operation. The enclosing lines comprise the pure basic block.

We identify pure basic block inputs and outputs as values in relevant registers upon input and upon exit. Free registers in the basic block are registers that are read from before they are assigned to, and are considered inputs to the pure basic block. Values in the final registers of aligned pure basic blocks are considered the outputs of the pure basic block.

For pure basic blocks with global references, semantics of the referenced entities are extracted from the full program sequence by performing a string-matching search for the referenced label and its following definition.

**Translating pure basic blocks.**    We lift assembly blocks from their corresponding hardware languages into an intermediate form usable by the synthesis engine. In this work, pure basic blocks that may be marked as potentially erroneous can be marked due to either global references or low-confidence token predictions.

Potential errors due to global references are solved using a custom solver designed for resolving global references. Pure basic blocks with global references must include the definition of the referenced entity in its semantics. The aligned entity on the input side, whether retrieved from its global definition or directly obtained from the input pure basic block, is translated into its bitvector representation. The pure basic block sequence and the bitvector representation of the correct entity value are passed to the global reference solver.

Potential errors due to low-confidence token predictions are solved using the Rosette (Torlak & Bodik, 2013) program synthesis engine. Aligned input and output sketch subsequences $x_{p_x}$ and $s$ are lifted into Rosette functions $P_{x_{p_x}}$ and $P_s$, where $P_s$ is a partial program with holes replaced by Rosette symbolic constants. The lifting is done by mapping each assembly line to its Rosette counterpart according to the semantics of the corresponding assembly hardware ISA.

**Solving the sketch.**    The global reference solver solves for hole mappings in output pure basic block sketch by either resolving the global reference label used or directly translating the entity

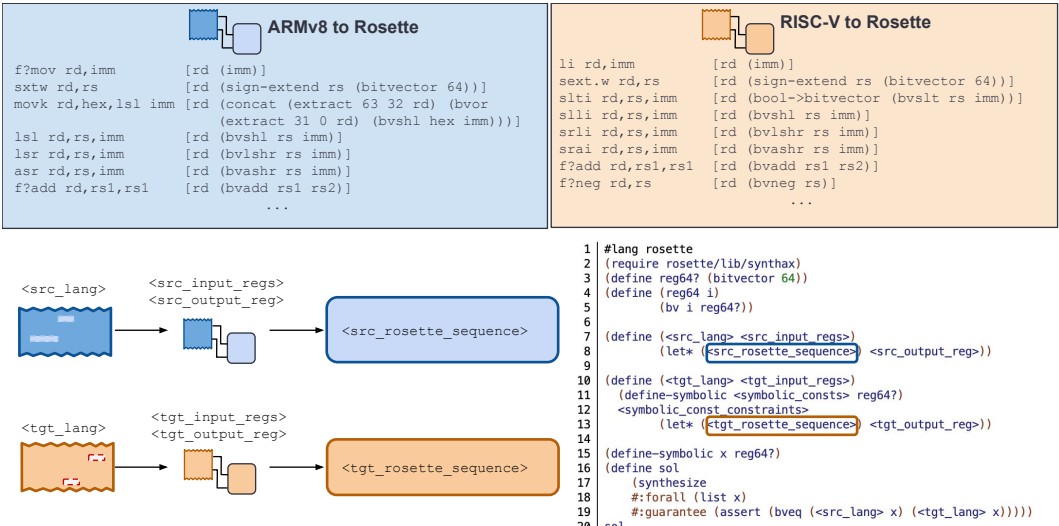

Figure 5: Assembly instructions are mapped to Rosette instructions according to the semantics of the corresponding assembly hardware ISA (sample shown at top). Holes in the sequence (indicated in dashed red rectangles) are translated into Rosette symbolic constants. The resulting Rosette instructions, along with the input and output registers, are plugged into a Rosette function template (bottom) to generate a full Rosette program whose solution produces a corrected mapping from holes to values.

| Model (# params) | L.R. | Batch | No. Steps | LoRA $r$ | LoRA Modules | Quant. |
|---|---|---|---|---|---|---|
| Enc-Decoder (400M) | 3e-5 | 8 | 520k | - | - | - |
| Starcoder-Base (15.5B) | 5e-6 | 16 | 2.9k | 16 | c_proj,c_attn,q_attn | int8 |
| CodeLlama (13B) | 5e-6 | 16 | 2.9k | 16 | q_proj,v_proj | int8 |

Table 4: Training details for language models used.

in the block. If the erroneous token in the output pure basic block is a reference label, the solver searches for entity definitions in the full generated program sequence whose bitvector representation matches the desired bitvector value set by the input sequence. If it finds a match, the label of the identified definition replaces the hole left in the sketch. If the solver does not find a match, it creates a new global definition with a unique label, and uses that label to replace the hole left in the sketch. If the erroneous token in the output pure basic block is a numerical value, the solver translates the desired bitvector value set by the input sequence into the representation expected by the ISA and replaces the hole left in the sketch with the resulting value.

Sketches for errors due to low-confidence tokens are solved by Rosette. Rosette solves for the hole mappings by ensuring that for all program inputs, the two functions are equivalent. This process is shown in Figure 5.

## A.2 MODEL TRAINING DETAILS

Details about training the generative language models are shared in Table 4

## B ADDITIONAL EXPERIMENTS

To further test the benefit of GUESS & SKETCH over just the language model approach, we run experiments with more Project Euler examples. We collect solutions to 82 additional unique Project Euler problems implemented in C [10], and compile them to the ARMv8 and RISC-V ISAs under

---

[10]https://github.com/LaurentMazare/ProjectEuler/tree/master

| Method | RISC-V to ARMv8 | ARMv8 to RISC-V |
|---|---|---|
| Encoder-Decoder | 34.1% | 37.8% |
| **GUESS & SKETCH** | **41.5%** | **51.2%** |

Table 5: Performance on More Project Euler problems.

the `-O0` optimization flag. The average number of lines in these programs is 246. The results of running the strongest baseline and our method are shown in Table 5. GUESS & SKETCH continues to provide performance gains averaging approximately 10%.

## C  CATEGORIZATION OF FAILED TRANSPILATIONS

Failed transpilations are categorized under one of several bottleneck failure reasons, listed in order of precedence. Process failures include length and process failure, in which the very process of transpilation fails on the given input. If an example does not encounter process failure, the next category is compilation failures including using the incorrect ISA instructions or global references. If the example successfully compiles, the next category of failures it may encounter is semantic failures including mathematic reasoning, copying, operational logic, and memory mis-management. These categories are further described below.

**Length.**   Some transpilation methods suffer from long input and output sequences. For example, current attention-based language models generally have a context window limit, so sequences that exceed that context window length will not be able to be processed by the language model.

**Process failure.**   Examples that fall under this category are ones where the transpilation process fails when processing the input, such as the rules-based transpiler that breaks down upon receiving an input that it cannot parse.

**Incorrect ISA.**   In assembly transpilation, the produced sequences must use exactly the instructions and entities available to the hardware in question. Failure examples that fall under this category produce sequences mistakenly use syntax that is incorrect or that actually belongs to a different ISA.

**Global references.**   Assembly programs might make references to entities that are invalid, or otherwise use or define global reference labels incorrectly. In these cases, the program will fail.

**Mathematic.**   Math errors are ones in which the translation process fails to correctly perform the required mathematic reasoning for a translation. Examples include translating code idioms such as different implementations of division (Möller & Granlund, 2011), addition and subtraction of large constants, and translation of float values to their IEEE 754 representations (iee, 1985).

**Copying.**   Copying errors are ones in which part of the input sequence fails to be copied to the output sequence. Examples include copying of constant strings, constant numeric values, and custom function names.

**Incorrect operation or register logic.**   The produced assembly sequence may use syntactically valid but semantically incorrect logic. These logical errors involve incorrect register or operation use, and the subsequent propagation of such mistakes.

**Memory mis-management.**   Assembly code must be able to reason about values in memory and manage memory access. Errors in this category are indicated by attempts to access memory at incorrect or invalid stack or memory locations, which may yield stash smashing, stack overflow, or segmentation faults in the latter, and unexpected values in either case.

### C.1  EXAMPLE ERRONEOUS TRANSPILATIONS

In this section, we include more example erroneous transpilations from different methods.

**LLM output: RISC-V**

```
            ...

    ...                       main:
addi  a4,s0,-40                 addi  sp,sp,-8272
lbu   a5,0(a4+a5)               sd    ra,8264(sp)
sub   a5,a5,64                  sd    r0,8256(sp)
lw    a4,-64(s0)               sd    s1,8248(sp)
                                addi  s0,sp,8272
    ...                             ...
```

| (a) Arguments are invalid. | (b) Offset values are out of range. |
| --- | --- |

Figure 6: The fine-tuned pre-trained code models tend to use instructions from ISAs other than the one which it is directed to use. Underlined arguments indicate invalid productions.

**Mistakes from fine-tuned code LLMs.** Pre-trained code language models, even after fine-tuning with examples in domain, tend to make more ISA mistakes than do other methods. Figure 6 shows two examples of erroneous generated code from a fine-tuned Starcoder-Base method. Figure 6a shows an example of the fine-tuned Starcoder-Base method producing code that is largely correct, but violates syntactic rules of the target hardware (RISC-V) by using added-register offsets for the `lbu` instructions. The syntax of RISC-V 64 does not allow register value addition for loading unsigned bytes by address. It also only allows subtraction by a specified register value rather than an immediate. Figure 6b shows code that allocates then uses a large stack space, but in doing so actually violates syntactic rules of the target hardware (RISC-V) by using an immediate value outside the legal 12-bit immediate ranges for the `addi` and `sd` instructions.

## D  BASELINE IMPLEMENTATION DETAILS

### D.1  PROMPTING GPT-4

The prompt used to extract translations from GPT-4 for Arm to RISC-V is as follows. For function translations:

You are able to translate assembly code from ARMv8 to RISC-V 64.

ARMv8:
main:\n.LFB0:\n\t.cfi_startproc\n\tstp\tx29, x30, [sp, -48]!\n\t.cfi_def_cfa_offset 48\n\t.cfi_offset 29, -48\n\t.cfi_offset 30, -40\n\tmov\tx29, sp\n\tadrp\tx0, :got:__stack_chk_guard\n\tldr\tx0, [x0, #:got_lo12:__stack_chk_guard]\n\tldr\tx1, [x0]\n\tstr\tx1, [sp, 40]\n\tmov\tx1, 0\n\tadrp\tx0, .LC0\n\tadd\tx0, x0, :lo12:.LC0\n\tbl\tprintf\n\tadd\tx0, sp, 24\n\tmov\tx1, x0\n\tadrp\tx0, .LC1\n\tadd\tx0, x0, :lo12:.LC1\n\tbl\t__isoc99_scanf\n\tldr\tw0, [sp, 24]\n\tmov\tw1, 34953\n\tmovk\tw1, 0x8888, lsl 16\n\tsmull\tx1, w0, w1\n\tlsr\tx1, x1, 32\n\tadd\tw1, w0, w1\n\tasr\tw1, w1, 4\n\tasr\tw0, w0, 31\n\tsub\tw1, w1, w0\n\tmov\tw0, 1500\n\tmul\tw0, w1, w0\n\tstr\tw0, [sp, 28]\n\tldr\tw1, [sp, 24]\n\tmov\tw0, 34953\n\tmovk\tw0, 0x8888, lsl 16\n\tsmull\tx0, w1, w0\n\tlsr\tx0, x0, 32\n\tadd\tw0, w1, w0\n\tasr\tw2, w0, 4\n\tasr\tw0, w1, 31\n\tsub\tw2, w2, w0\n\tmov\tw0, w2\n\tlsl\tw0, w0, 4\n\tsub\tw0, w0, w2\n\tlsl\tw0, w0, 1\n\tsub\tw2, w1, w0\n\tmov\tw0, w2\n\tlsl\tw0, w0, 2\n\tadd\tw0, w0, w2\n\tlsl\tw0, w0, 3\n\tstr\tw0, [sp, 32]\n\tldr\tw1, [sp, 28]\n\tldr\tw0, [sp, 32]\n\tadd\tw0, w1, w0\n\tstr\tw0, [sp, 36]\n\tldr\tw1, [sp, 36]\n\tadrp\tx0, .LC2\n\tadd\tx0, x0, :lo12:.LC2\n\tbl\tprintf\n\tmov\tw0, 0\n\tmov\tw1, w0\n\tadrp\tx0, :got:__stack_chk_guard\n\tldr\tx0, [x0, #:got_lo12:__stack_chk_guard]\n\tldr\tx3, [sp, 40]\n\tldr\tx2, [x0]\n\tsubs\tx3, x3, x2\n\tmov\tx2, 0\n\tbeq\t.L3\n\tbl\t__stack_chk_fail\n.L3:\n\tmov\tw0, w1\n\tldp\tx29, x30, [sp], 48\n\t.cfi_restore 30\n\t.cfi_restore 29\n\t.cfi_def_cfa_offset 0\n\tret\n\t.cfi_endproc\n

RISC−V 64:
main:\n\taddi\tsp,sp,−48\n\tsd\tra,40(sp)\n\tsd\ts0,32(sp)\n\taddi\ts0,sp
,48\n\tla\ta5,__stack_chk_guard\n\tld\ta4,0(a5)\n\tsd\ta4,−24(s0)\n
\tli\ta4,0\n\tlla\ta0,.LC0\n\tcall\tprintf@plt\n\taddi\ta5,s0,−40\n\
tmv\ta1,a5\n\tlla\ta0,.LC1\n\tcall\t__isoc99_scanf@plt\n\tlw\ta5,−40(
s0)\n\tmv\ta4,a5\n\tli\ta5,30\n\tdivw\ta5,a4,a5\n\tsext.w\ta4,a5\n\
tli\ta5,1500\n\tmulw\ta5,a4,a5\n\tsw\ta5,−36(s0)\n\tlw\ta5,−40(s0)\n\
tmv\ta4,a5\n\tli\ta5,30\n\tremw\ta5,a4,a5\n\tsext.w\ta5,a5\n\tmv\ta4,
a5\n\tmv\ta5,a4\n\tslliw\ta5,a5,2\n\taddw\ta5,a5,a4\n\tslliw\ta5,a5
,3\n\tsw\ta5,−32(s0)\n\tlw\ta5,−36(s0)\n\tmv\ta4,a5\n\tlw\ta5,−32(s0)
\n\taddw\ta4,a5\n\tsw\ta5,−28(s0)\n\tlw\ta5,−28(s0)\n\tmv\ta1,a5\n
\tlla\ta0,.LC2\n\tcall\tprintf@plt\n\tli\ta5,0\n\tmv\ta4,a5\n\tla\ta5
,__stack_chk_guard\n\tld\ta3,−24(s0)\n\tld\ta5,0(a5)\n\txor\ta5,a3
,a5\n\tli\ta3,0\n\tbeq\ta5,zero,.L3\n\tcall\t__stack_chk_fail@plt\n
.L3:\n\tmv\ta0,a4\n\tld\tra,40(sp)\n\tld\ts0,32(sp)\n\taddi\tsp,sp
,48\n\tjr\tra\n

ARMv8:
main:\n.LFB6:\n\t.cfi_startproc\n\tstp\tx29, x30, [sp, −64]!\n\t.
cfi_def_cfa_offset 64\n\t.cfi_offset 29, −64\n\t.cfi_offset 30, −56\n
\tmov\tx29, sp\n\tadrp\tx0, :got:__stack_chk_guard\n\tldr\tx0, [x0,
#:got_lo12:__stack_chk_guard]\n\tldr\tx1, [x0]\n\tstr\tx1, [sp, 56]\n
\tmov\tx1, 0\n\tadrp\tx0, .LC0\n\tadd\tx0, x0, :lo12:.LC0\n\tbl\
tprintf\n\tadd\tx0, sp, 20\n\tmov\tx1, x0\n\tadrp\tx0, .LC1\n\tadd\
tx0, x0, :lo12:.LC1\n\tbl\t__isoc99_scanf\n\tldr\tw0, [sp, 20]\n\tmov
\tw1, w0\n\tadrp\tx0, .LC2\n\tadd\tx0, x0, :lo12:.LC2\n\tbl\tprintf\n
\tadrp\tx0, .LC3\n\tadd\tx0, x0, :lo12:.LC3\n\tbl\tprintf\n\tadd\tx0,
sp, 19\n\tmov\tx1, x0\n\tadrp\tx0, .LC4\n\tadd\tx0, x0, :lo12:.LC4\n
\tbl\t__isoc99_scanf\n\tldrb\tw0, [sp, 19]\n\tmov\tw1, w0\n\tadrp\tx0
, .LC5\n\tadd\tx0, x0, :lo12:.LC5\n\tbl\tprintf\n\tadrp\tx0, .LC6\n\
tadd\tx0, x0, :lo12:.LC6\n\tbl\tprintf\n\tadd\tx0, sp, 24\n\tmov\tx1,
 x0\n\tadrp\tx0, .LC7\n\tadd\tx0, x0, :lo12:.LC7\n\tbl\
t__isoc99_scanf\n\tldr\td0, [sp, 24]\n\tadrp\tx0, .LC8\n\tadd\tx0, x0
, :lo12:.LC8\n\tbl\tprintf\n\tadrp\tx0, .LC9\n\tadd\tx0, x0, :lo12:.
LC9\n\tbl\tprintf\n\tadrp\tx0, :got:stdin\n\tldr\tx0, [x0, #:got_lo12
:stdin]\n\tldr\tx1, [x0]\n\tadd\tx0, sp, 32\n\tmov\tx2, x1\n\tmov\tw1
, 20\n\tbl\tfgets\n\tadd\tx0, sp, 32\n\tmov\tx1, x0\n\tadrp\tx0, .
LC10\n\tadd\tx0, x0, :lo12:.LC10\n\tbl\tprintf\n\tmov\tw0, 0\n\tmov\
tw1, w0\n\tadrp\tx0, :got:__stack_chk_guard\n\tldr\tx0, [x0, #:
got_lo12:__stack_chk_guard]\n\tldr\tx3, [sp, 56]\n\tldr\tx2, [x0]\n\
tsubs\tx3, x3, x2\n\tmov\tx2, 0\n\tbeq\t.L3\n\tbl\t__stack_chk_fail\n
.L3:\n\tmov\tw0, w1\n\tldp\tx29, x30, [sp], 64\n\t.cfi_restore 30\n\t
.cfi_restore 29\n\t.cfi_def_cfa_offset 0\n\tret\n\t.cfi_endproc\n

RISC−V 64:
main:\n\taddi\tsp,sp,−64\n\tsd\tra,56(sp)\n\tsd\ts0,48(sp)\n\taddi\ts0,sp
,64\n\tla\ta5,__stack_chk_guard\n\tld\ta4,0(a5)\n\tsd\ta4,−24(s0)\n
\tli\ta4,0\n\tlla\ta0,.LC0\n\tcall\tprintf@plt\n\taddi\ta5,s0,−60\n\
tmv\ta1,a5\n\tlla\ta0,.LC1\n\tcall\t__isoc99_scanf@plt\n\tlw\ta5,−60(
s0)\n\tmv\ta1,a5\n\tlla\ta0,.LC2\n\tcall\tprintf@plt\n\tlla\ta0,.LC3\
n\tcall\tprintf@plt\n\taddi\ta5,s0,−61\n\tmv\ta1,a5\n\tlla\ta0,.LC4\n
\tcall\t__isoc99_scanf@plt\n\tlbu\ta5,−61(s0)\n\tsext.w\ta5,a5\n\tmv\
ta1,a5\n\tlla\ta0,.LC5\n\tcall\tprintf@plt\n\tlla\ta0,.LC6\n\tcall\
tprintf@plt\n\taddi\ta5,s0,−56\n\tmv\ta1,a5\n\tlla\ta0,.LC7\n\tcall\
t__isoc99_scanf@plt\n\tfld\tfa5,−56(s0)\n\tfmv.x.d\ta1,fa5\n\tlla\ta0
,.LC8\n\tcall\tprintf@plt\n\tlla\ta0,.LC9\n\tcall\tprintf@plt\n\tla\
ta5,stdin\n\tld\ta4,0(a5)\n\taddi\ta5,s0,−48\n\tmv\ta2,a4\n\tli\ta1
,20\n\tmv\ta0,a5\n\tcall\tfgets@plt\n\taddi\ta5,s0,−48\n\tmv\ta1,a5\n
\tlla\ta0,.LC10\n\tcall\tprintf@plt\n\tli\ta5,0\n\tmv\ta4,a5\n\tla\
ta5,__stack_chk_guard\n\tld\ta3,−24(s0)\n\tld\ta5,0(a5)\n\txor\ta5,
a3,a5\n\tli\ta3,0\n\tbeq\ta5,zero,.L3\n\tcall\
t__stack_chk_fail@plt\n.L3:\n\tmv\ta0,a4\n\tld\tra,56(sp)\n\tld\ts0
,48(sp)\n\taddi\tsp,sp,64\n\tjr\tra\n

ARMv8:

b:\n\t.zero\t8\n\t.global\tc\n\t.align\t3\n\t.type\tc, %object\n\t.size\tc, 8\n

RISC−V 64:
b:\n\t.zero\t8\n\t.globl\tc\n\t.align\t3\n\t.type\tc, @object\n\t.size\tc, 8\n

ARMv8:
foo:\n.LFB0:\n\t.cfi_startproc\n\tstp\tx29, x30, [sp, −16]!\n\t.cfi_def_cfa_offset 16\n\t.cfi_offset 29, −16\n\t.cfi_offset 30, −8\n\tmov\tx29, sp\n\tadrp\tx0, global\n\tadd\tx0, x0, :lo12:global\n\tbl\tbar\n\tadrp\tx0, global_2\n\tadd\tx0, x0, :lo12:global_2\n\tbl\tbar\n\tadrp\tx0, :got:global_3\n\tldr\tx0, [x0, #:got_lo12:global_3]\n\tbl\tbar\n\tadrp\tx0, global_5\n\tadd\tx0, x0, :lo12:global_5\n\tbl\tbar\n\tadrp\tx0, global_6\n\tadd\tx0, x0, :lo12:global_6\n\tbl\tbar\n\tnop\n\tldp\tx29, x30, [sp], 16\n\t.cfi_restore 30\n\t.cfi_restore 29\n\t.cfi_def_cfa_offset 0\n\tret\n\t.cfi_endproc\n

RISC−V 64:
foo:\n\taddi\tsp,sp,−16\n\tsd\tra,8(sp)\n\tsd\ts0,0(sp)\n\taddi\ts0,sp,16\n\tlla\ta0,global\n\tcall\tbar@plt\n\tlla\ta0,global_2\n\tcall\tbar@plt\n\tla\ta0,global_3\n\tcall\tbar@plt\n\tlla\ta0,global_5\n\tcall\tbar@plt\n\tlla\ta0,global_6\n\tcall\tbar@plt\n\tnop\n\tld\tra,8(sp)\n\tld\ts0,0(sp)\n\taddi\tsp,sp,16\n\tjr\tra\n

ARMv8:
{insert input code to translate}

RISC−V 64:

For outer file translations:

You are able to translate assembly code from ARMv8 to RISC−V 64.

ARMv8:
\t.arch armv8−a\n\t.file\t"program19928025.c"\n\t.text\n\t.section\t.rodata\n\t.align\t3\n.LC0:\n\t.string\t"Enter your age: "\n\t.align\t3\n.LC1:\n\t.string\t"%d"\n\t.align\t3\n.LC2:\n\t.string\t"You are %d years old.\\n"\n\t.align\t3\n.LC3:\n\t.string\t"Enter your grade: "\n\t.align\t3\n.LC4:\n\t.string\t"%c"\n\t.align\t3\n.LC5:\n\t.string\t"Your grade is: %c"\n\t.align\t3\n.LC6:\n\t.string\t"Enter your gpa: "\n\t.align\t3\n.LC7:\n\t.string\t"%lf"\n\t.align\t3\n.LC8:\n\t.string\t"Your gpa is: %lf \\n"\n\t.align\t3\n.LC9:\n\t.string\t"Enter your name: "\n\t.align\t3\n.LC10:\n\t.string\t"Your name is %s"\n\t.text\n\t.align\t2\n\t.global\tmain\n\t.type\tmain, %function\n{main}.LFE6:\n\t.size\tmain, .−main\n\t.ident\t"GCC: (Ubuntu 11.3.0−1ubuntu1~22.04) 11.3.0"\n\t.section\t.note.GNU−stack,"",@progbits\n

RISC−V 64:
\t.file\t"program19928025.c"\n\t.option pic\n\t.text\n\t.section\t.rodata\n\t.align\t3\n.LC0:\n\t.string\t"Enter your age: "\n\t.align\t3\n.LC1:\n\t.string\t"%d"\n\t.align\t3\n.LC2:\n\t.string\t"You are %d years old.\\n"\n\t.align\t3\n.LC3:\n\t.string\t"Enter your grade: "\n\t.align\t3\n.LC4:\n\t.string\t"%c"\n\t.align\t3\n.LC5:\n\t.string\t"Your grade is: %c"\n\t.align\t3\n.LC6:\n\t.string\t"Enter your gpa: "\n\t.align\t3\n.LC7:\n\t.string\t"%lf"\n\t.align\t3\n.LC8:\n\t.string\t"Your gpa is: %lf \\n"\n\t.align\t3\n.LC9:\n\t.string\t"Enter your name: "\n\t.align\t3\n.LC10:\n\t.string\t"Your name is %s"\n\t.text\n\t.align\t1\n\t.globl\tmain\n\t.type\tmain, @function\n{main}\t.size\tmain, .−main\n\t.ident\t"GCC: (Ubuntu 11.3.0−1ubuntu1~22.04) 11.3.0"\n\t.section\t.note.GNU−stack,"",@progbits\n

ARMv8:
\t.arch armv8−a\n\t.file\t"program12490936.c"\n\t.text\n\t.section\t.rodata\n\t.align\t3\n.LC0:\n\t.string\t"Enter the distance the van

has travelled:"\n\t.align\t3\n.LC1:\n\t.string\t"%d"\n\t.align\t3\n.LC2:\n\t.string\t"Amount = %d"\n\t.text\n\t.align\t2\n\t.global\tmain\n\t.type\tmain, %function\n{main}.LFE0:\n\t.size\tmain, .-main\n\t.ident\t"GCC: (Ubuntu 11.3.0-1ubuntu1~22.04) 11.3.0"\n\t.section\t.note.GNU-stack,"",@progbits\n

RISC-V 64:
\t.file\t"program12490936.c"\n\t.option pic\n\t.text\n\t.section\t.rodata\n\t.align\t3\n.LC0:\n\t.string\t"Enter the distance the van has travelled:"\n\t.align\t3\n.LC1:\n\t.string\t"%d"\n\t.align\t3\n.LC2:\n\t.string\t"Amount = %d"\n\t.text\n\t.align\t1\n\t.globl\tmain\n\t.type\tmain, @function\n{main}\t.size\tmain, .-main\n\t.ident\t"GCC: (Ubuntu 11.3.0-1ubuntu1~22.04) 11.3.0"\n\t.section\t.note.GNU-stack,"",@progbits\n

ARMv8:
\t.arch armv8-a\n\t.file\t"program14079072.c"\n\t.text\n\t.global\tb\n\t.bss\n\t.align\t3\n\t.type\tb, %object\n\t.size\tb, 8\n{b}{c}{d}{e}{f}.LFE0:\n\t.size\tf, .-f\n\t.ident\t"GCC: (Ubuntu 11.3.0-1ubuntu1~22.04) 11.3.0"\n\t.section\t.note.GNU-stack,"",@progbits\n

RISC-V 64:
\t.file\t"program14079072.c"\n\t.option pic\n\t.text\n\t.globl\tb\n\t.bss\n\t.align\t3\n\t.type\tb, @object\n\t.size\tb, 8\n{b}{c}{d}{e}{f}\t.size\tf, .-f\n\t.ident\t"GCC: (Ubuntu 11.3.0-1ubuntu1~22.04) 11.3.0"\n\t.section\t.note.GNU-stack,"",@progbits\n

ARMv8:
\t.arch armv8-a\n\t.file\t"program17748089.c"\n\t.text\n\t.section\t.rodata\n\t.align\t3\n.LC0:\n\t.string\t"%f\\n%f\\n%f"\n\t.align\t3\n.LC1:\n\t.string\t"%lf"\n\t.text\n\t.align\t2\n\t.global\tmain\n\t.type\tmain, %function\n{main}.LFE0:\n\t.size\tmain, .-main\n\t.ident\t"GCC: (Ubuntu 11.3.0-1ubuntu1~22.04) 11.3.0"\n\t.section\t.note.GNU-stack,"",@progbits\n

RISC-V 64:
\t.file\t"program17748089.c"\n\t.option pic\n\t.text\n\t.section\t.rodata\n\t.align\t3\n.LC0:\n\t.string\t"%f\\n%f\\n%f"\n\t.align\t3\n.LC1:\n\t.string\t"%lf"\n\t.text\n\t.align\t1\n\t.globl\tmain\n\t.type\tmain, @function\n{main}\t.size\tmain, .-main\n\t.ident\t"GCC: (Ubuntu 11.3.0-1ubuntu1~22.04) 11.3.0"\n\t.section\t.note.GNU-stack,"",@progbits\n

ARMv8:
{insert input code to translate}

RISC-V 64:

The reverse direction reverses source and target language specifications accordingly.

