# OpenReview forum: "Guess & Sketch: Language Model Guided Transpilation"
_ICLR.cc/2024/Conference — ICLR 2024 poster_

### Official Review · Reviewer_LBov · 2023-11-01

**Soundness:** 3 good
**Presentation:** 4 excellent
**Contribution:** 3 good
**Rating:** 8
**Confidence:** 4

**Summary:**

this work presents a way of transpilation: turning one assembly code into another functionally equivalent assembly code.

the main techniques consists of: using a LM to generate candidate programs. from the internal values of the LM 1) alignment/attention and 2) uncertainty, generate localized sketches with holes for the candidate program. this localized sketch is then solved, with the holes resolved to values that are provably equivalent to the source code's corresponding fragments. the fragments are then stitched together, finishing the transpilation process.

results show the proposed method beats a reasonable set of baselines -- a heuristic based transpiler, and few-shot using gpt4.

**Strengths:**

## the good part of quality: that it worked
The presented method works, on a domain of highly structured translation task (i.e. highly stylized texts), something a language model should perform very well at, and it shows. The extra care taken to correct the translation locally is a reasonable yet good idea to complement the weakness of the language model.

The benchmark is thorough, and the evaluation (on what is being shown) is solid.

## clarity
I am very grateful how this work is able to encapsulate the domain specific aspects of compiler and assembly, so that the top level algorithm remains accessible to the ML audience. Thank you!

## novelty : fair
I think it is a straight forward paper, and it outlined reasonable decompositions of the transpiling tasks to LLM and a symbolic solver.

**Weaknesses:**

## the not so good part of quality:

### evaluation set is small
This work can be significantly beefed up with a synthetic test set. Evaluation on mere 100s of programs is likely not sufficient. Since it is possible to compile C into both architectures, and since test generation / fuzzing is a well established approach, this work can benefit from an artificial/synthetic test set consists of about ~1k programs, to evaluate the correctness of the transpiler more thoroughly.

### lack of statistic tests
At least we should see confidence intervals of the results, or some kind of t-test to make sure that the proposes method is better than the baseline not due to noise. Kindly ask your colleagues in statistics to look over your tables and give recommendations on how it could be made bullet proof.

I would love to see this update in the rebuttal period.

## fit of venue
While I think this is a good paper, it might be a better fit at PLDI. As I am unsure what is the AI/ML lessons gained from this work, other than it is possible to build such a system, and some relatively detailed finding on how well language models are at learning over a highly stylized text (assembly code) when compared to English sentences.

However, as other application papers of the compiler flavour has a precedence of appearing at ICLR, this is not a major concern.

**Questions:**

## program equivalence?

As I understand program equivalence is an undecidable problem. If I recall correctly, synthesis systems like sketch does not have a way to fully verify the SKETCH and the SPEC are identical over all input-outputs, but only over a bounded domain?

Is this an issue for your work? Or is it because everything is bounded to begin with, as we're working over assembly and we only need to account for, for instance, 16 bits integers or similar ? Or is it some Z3 theory magic that allows for a DSL which programs can be reasoned for full equivalence?

At any rate, this should probably be clarified in the paper.

## successfully compile = ?

If I read correctly, success is measured over a set of input-output test cases to see if the input code runs the same as the compiled output code. Is this related to the program equivalence problem above somehow? Is this comprehensive enough to make sure the transpiling is not mistaken?

---

> ### Author Response · Authors · 2023-11-17
>
> Thank you for your careful consideration and detailed review of GUESS & SKETCH. The questions and suggestions about evaluation have been illuminating, and we respond to them below.
>
> **Re: evaluation set is small.**
>
> Thanks for the feedback.
>
> To further rigorously evaluate the correctness of the GUESS & SKETCH transpiler against the most competitive transpiler baseline, we collected an additional test set of 82 real-world programs, and conducted experiments in both translation directions. The results show that our method correctly transpiles 41.5% of programs from RISC to ARM, compared to 34.1% by the Encoder-Decoder, and 51.2% from ARM to RISC, compared to 37.8% by the Encoder-Decoder. Details are further shared in the Appendix, section B.
>
> We note that we were limited by the goal of collecting high quality, evaluable programs. The hundreds of programs used in the existing test sets are collected from real mathematic programming challenge problems, computer program performance benchmarks, and realistic interactive programs.
>
> **Re: lack of statistical tests**
>
> Thanks for the suggestion.
>
> We ran statistical tests on our results. The top two competitive baselines are the engineered Transpiler and the Encoder-Decoder. The z-score against the Transpiler is 6.89 and against the Encoder-Decoder is 4.57, both of which allow us to conclude with a significance level of 0.01 that GUESS & SKETCH outperforms the baselines. While we agree that thousands of synthetic programs could provide additional support for comparing the transpilation methods, the existing results do show strong statistical significance.
>
> **Re: fit of venue**
>
> We agree that cross-disciplinary nature of this work makes it a suitable paper for multiple different venues. We note previous ICLR submissions that explored code generation/translation (Synchromesh, Poesy et al. 2022; Leveraging Automated Unit Tests for Unsupervised Code Translation, Rosier et al. 2022; InCoder, Fried et al. 2023; Binding Language Models in Symbolic Languages, Cheng et al. 2023; DocPrompting, Zhou et al. 2023; etc.) and compiler innovation (DLVM, Wei et al. 2018; Code Translation with Compiler Representations, Szafraniec et al. 2023; etc.) have also been well-received.  Given the machine learning and program synthesis backbone of GUESS & SKETCH, as well as the real-world application of the task proposed, we think that many members of the ICLR audience would be interested in reading and building on this work.
>
> **Re: measuring success and “program equivalence”**
>
> We use test case input-output equivalence to measure success in the transpilation task. This is not a true measure for program equivalence, but this evaluation setup has been used as a proxy for success in related works in inductive program synthesis (Flashfill, Gulwani 2011; DreamCoder, Ellis et al. 2020) and today's major code generation benchmarks (HumanEval, Chen et al. 2021; MBPP, Austin et al. 2021; APPS, Hendricks et al. 2021).
>
> **Re: verification of subprogram equivalence in Sketch**
>
> Yes you are correct that this domain needs to be specified for Sketch. We fix the domain of inputs over which Sketch verifies equality. This input domain can be restricted for this domain, specifically by the registers of the assembly architectures used, which in this work is size-64 bitvectors.

---

> > ### Comment · Reviewer_LBov · 2023-11-23
> > **substantial response that warrants a raised score**
> >
> > thanks for addressing these concerns, I'm raising my score to 7, but since there is no 7, I rounded up to 8.
> > I do think this work is substantial enough and thorough enough after the updates.

---

### Official Review · Reviewer_awtu · 2023-11-01

**Soundness:** 4 excellent
**Presentation:** 4 excellent
**Contribution:** 4 excellent
**Rating:** 8
**Confidence:** 2

**Summary:**

This paper presents an approach for translating low-level assembly programs into higher-level code for the purpose of analysis and human understanding. The approach is based on a combination of neural processing and symbolic program translation. The proposed approach,  called GUESS & SKETCH, extracts alignment information from features of the neural language model and passes it to a symbolic solver to perform "transpilation". The paper also presents experiments illustrating the benefits of GUESS & SKETCH as compared to GPT-4 and an engineered "transpiler".

**Strengths:**

The paper is well written and presents a clear contribution. The combination of generative language models and program synthesis by sketching is new and it is shown to be effective as compared to state of the art techniques.

**Weaknesses:**

I could not understand the correctness guarantee provided by the approach. The authors say "the correctness of GUESS & SKETCH is always lower-bounded by the correctness of the initial guess" -- the authors should explain what they mean by lower bound here. If the translation is incorrect, how can it be useful in practice?

The scalability is unclear.  What s the largest program that has been translated using the approach presented here?

**Questions:**

Please see above?

---

> ### Author Response · Authors · 2023-11-17
>
> We thank the reviewer for their comments. The questions raised highlight important clarifications about the benefits of our approach.
>
> **Re: correctness lower bound**
>
> Let us clarify this statement, as we did not mean to overclaim here. What we mean by “the correctness of GUESS & SKETCH is always lower-bounded by the correctness of the initial guess” is that *if* the initial guess generated by the language model is correct, then GUESS & SKETCH does nothing additional and simply returns the correct guess. If the original generation is incorrect, GUESS & SKETCH may or may not be able to correct it. Therefore, GUESS & SKETCH is always lower-bounded in correctness by the correctness of the language model. Note that GUESS & SKETCH, and no existing neural approach, *guarantee* correctness of the translation.
>
> **Re: downstream use of incorrect translation**
>
> An incorrect translation may not be directly useful as-is. Subsequent post-processing may be used to correct an incorrect translation. In this work, we show how an automated solver-based post-processing step (SKETCH) can perform corrections. Other future work may examine other methods of correction, such as with hints to humans, synthesis engines, or other language models.
>
> **Re: scalability**
>
> GUESS & SKETCH is limited to *functions* that are shorter than the context window of the underlying language model. The context window of the language model used in this paper is 2048 tokens, but could be exteneded. Since multiple functions compose a full program, the program can be arbritrarily long. The largest program we successfully transpile is fannkuch-redux-3 in the Benchmarks dataset. It has 10 functions and 1 header. In ARM, this is 4297 tokens total (longest function has 1405 tokens). In RISC, this is 3877 tokens total (longest function has 1321 tokens).

---

> > ### Comment · Reviewer_awtu · 2023-11-21
> >
> > Thank you for the clarifications. I maintain my score.

---

> > > ### Author Response · Authors · 2023-11-22
> > >
> > > Thank you!

---

### Official Review · Reviewer_nDKt · 2023-11-01

**Soundness:** 3 good
**Presentation:** 4 excellent
**Contribution:** 2 fair
**Rating:** 5
**Confidence:** 2

**Summary:**

The paper presents an approach for machine language translation. They attempt to utilise a generative language model with a confidence score to identify uncertain blocks or "guesses" which can then be symbolically solved using a neuro-symbolic solver. They rely on Sketch (Solar-Lexama  et al)  prior work for handling the expansion/completion of the uncertain tokens. The authors perform experiments on other three datasets (Unix, Euler, Benchmarks) outperforming or equal (in rare cases) in all test settings.

**Strengths:**

- While a simple concept, the method outperforms prior work
- The concept of uncertainty is a good mapping to identify holes in the generated program
- Evaluation is robust and thorough, providing analysis of failure cases
- Authors identify a setting for Neuro-symbolic approaches to work stably and outperform prior works

**Weaknesses:**

- Novelty within this approach is quite limited the translation is a standard approach the confidence is simple (see below), and they use an existing neuro-symbolic solver therefore, it is more on the sole idea of putting these together. This is the main criticism. However, they outperform prior work, and the idea is interesting and technically sound.
- Confidence is very trivially explained. In general, deep models are very confident even when they are wrong. It isn't clear how this was implemented and is critical to the method. As the author's rely on this to identify potential errors for solving.
- The parameter lambda is not ablated on as the threshold for identifying blocks. It is unclear if this is set low to allow more errors i.e. false negatives but to make the result more robust.

**Questions:**

- Explain more how the confidence is applied and used is it based on prior work as there is significant literature in this area
- Does the Lamda hyper-parameter effect the output, was this ablated on, but not included?
- Why do you choose only the region of error to be solved? Did you consider using a buffer before and after as well to increase the consistency across the section and provide context?

---

> ### Author Response · Authors · 2023-11-17
>
> Thank you for your insights and questions. We have incorporated feedback to clarify how we use model prediction confidence and determine regions of error to correct. Please find below detailed responses and updated draft to address the listed questions.
>
> **Re: novelty**
>
> We agree that the final version of GUESS & SKETCH is quite simple. We experimented with more complex methods, but were able to simplify things down to a few core ideas (model confidence → sketch solver). We believe that this simplicity is an advantage for using this approach in practice.
>
> **Re: model prediction confidence. is this an imperfect measure of mistakes**
>
> Our goal of identifying potential mistakes is to filter which output subsequences to correct, since correcting every output subsequences is too computationally expensive. Indeed, as previous works have studied (R-U-Sure, Johnson et al. 2023, 'Generation Probabilities Are Not Enough', Vasconcelos et al. 2023), the underlying language model can be very confident about incorrect predictions, so confidence alone may not be sufficient to identify all errors. However, empirically in this domain we found that there was significant signal in this property, as shown by the experimental results. While neither confidence nor attention provide hard guarentees, they both provide signal in this domain. Future versions of GUESS & SKETCH can be further improved by devising better domain-general heuristics for identifying potential errors.
>
> **Re: ablation on gamma hyper-parameter**
>
> Thanks for the suggestion.
>
> We ran an ablation study on the gamma hyperparameter with the Project Euler test set. We found that a range of gamma hyper parameters still produces the same results. We sweep across gamma values: 0.8, 0.9, and 0.95 to study this trade-off between computation time and correctness. We find that across these gamma values, the number of corrected programs is the same, but the inference runtime increases with gamma. From 0.8 to 0.95, the inference time increases by 2.2x. This suggests that the model is extremely confident on most tokens, and that even a low-gamma is able to find plausible errors.
>
> **Re: how do we choose the region of error to be solved?**
>
> The region of error to be solved must be a contiguous subsequence whose source semantics can be extracted for the SKETCH step. In the assembly domain, we exploit the domain specific idea of a “pure basic blocks”. This is based on standard basic blocks (Computer Architecture, Patterson & Hennessy) and described further in Appendix A.1. For this problem and SKETCH setup, these block are functionally independent, so we would not improve by larger regions. For other domains we agree that we might need to expand the subsequence could provide more context.

---

### Author Response · Authors · 2023-11-17

We appreciate the thoroughness with which the reviewers read our paper, and the insightful questions raised in response. GUESS & SKETCH transpiles assembly programs with a neurosymbolic approach that outperforms all other existing approaches by leveraging the scalability of language model with the correctness assurances of symbolic methods.

The high-level questions were about: (1) measure of success in this task, (2) correctness guarantees, (3) selection of the confidence threshold hyper parameter (gamma), and (4) statistical significance of our evaluation test sets. Implementation questions interrogate (5) the selected region of error for correction, (6) scalability, and (7) input domain of the Sketch solver. Our individual author responses address each of these concerns and all other questions in detail. We have also updated the paper accordingly. In summary, our responses are as follows:

**(1) measure of success in this task** (LBov, awtu)**:** Program equivalence remains an unsolved problem, so, following suit of other program generation / translation tasks from both the machine learning (HumanEval, Chen et al. 2021; MBPP, Austin et al. 2021; APPS, Hendricks et al. 2021) and program synthesis (Flashfill, Gulwani 2011; DreamCoder, Ellis et al. 2020) fields, we use input-output equivalence as a proxy for program equivalence.

**(2) correctness guarantees** (awtu)**:** GUESS & SKETCH, like other existing neural approaches, does not guarantee correctness of the translation. However, GUESS & SKETCH is guaranteed to be at least as correct as the underlying language model. This is because if the original language model guess is correct, GUESS & SKETCH simply returns the original guess and does not propose any additional changes.

**(3) selection of gamma** (nDKt):  To evaluate the role of confidence as a filter for which basic blocks to correct, we run an additional ablation study on the gamma hyper parameter and find that across a range of confidence threshold hyper parameters (0.8-0.95), GUESS & SKETCH still produces the same transpilation success rate. This indicates that while confidence is not always indicative of errors, it is useful as a signal for determining where to apply sketch.

**(4) statistical significance of results** (LBov): We ran a two proportion z-test and confirm with a significance level of 0.01 that our method outperforms all other baselines. We include these in the updated version.

**(5) subsequence selection** (nDKt): We selected the region to be corrected so that we could construct a clear equivalence relation between the source and output aligned subsequences. We detail these subsequences as “pure basic blocks” in Appendix A.1: sequences of assembly code lines that have a single entry point, a single exit point, and no memory or stack management within the code sequence. This constrains pure basic blocks to be code sequences in which all data is either passed in via values already loaded into registers, or constant values coded into the sequence.

**(6) scalability** (awtu): GUESS & SKETCH translates each function of the program using a language model, so it is limited by the context window of the model. We use a context window of 2048 tokens in this paper. Since multiple functions compose a full program, in practice the program can be much longer. The largest program we successfully transpile is 4297 tokens in ARM and 3877 tokens in RISC.

**(7) Sketch input domain** (LBov): We constrain the domain of inputs to 64-bit bitvectors, consistent with the registers of the ARMv8 and RISC-V architectures we use. The underlying SKETCH solver reasons with full equivalence over this restricted input space.

We have incorporated the reviewer suggestions into our paper. Thank you all for your questions and insights!

---

### Meta-Review · Area_Chair_Nu76 · 2023-12-03

**Metareview:**

This paper presents GUESS & SKETCH, an approach that leverages LMs and symbolic solvers in a neurosymbolic approach to learned transpilation for assembly code, i.e., for turning low-level code into high-level, human-understandable code. Overall, as the reviews point out, this is very interesting and solid work. Indeed, the main contribution is in the clever and creative combination of different existing methods and applying it to an important problem.

**Justification For Why Not Higher Score:**

The contribution is not so much on the methodological levelr

**Justification For Why Not Lower Score:**

The vision of LM-guided transpilation is very interesting an creative.

---

### Decision · Program_Chairs · 2024-01-16

Accept (poster)